# Feature Screening for High-Dimensional Variable Selection in Generalized Linear Models

**DOI:** 10.3390/e25060851

**Published:** 2023-05-26

**Authors:** Jinzhu Jiang, Junfeng Shang

**Affiliations:** Department of Mathematics and Statistics, Bowling Green State University, Bowling Green, OH 43403, USA

**Keywords:** feature screening, high dimensional data, generalized linear models, logit model

## Abstract

The two-stage feature screening method for linear models applies dimension reduction at first stage to screen out nuisance features and dramatically reduce the dimension to a moderate size; at the second stage, penalized methods such as LASSO and SCAD could be applied for feature selection. A majority of subsequent works on the sure independent screening methods have focused mainly on the linear model. This motivates us to extend the independence screening method to generalized linear models, and particularly with binary response by using the point-biserial correlation. We develop a two-stage feature screening method called point-biserial sure independence screening (PB-SIS) for high-dimensional generalized linear models, aiming for high selection accuracy and low computational cost. We demonstrate that PB-SIS is a feature screening method with high efficiency. The PB-SIS method possesses the sure independence property under certain regularity conditions. A set of simulation studies are conducted and confirm the sure independence property and the accuracy and efficiency of PB-SIS. Finally we apply PB-SIS to one real data example to show its effectiveness.

## 1. Introduction

As the data with a huge number of features becomes popular in real life, many feature screening approaches have been developed to reduce the size of features [1]. introduced a model-free category-adaptive feature screening approach to detect category-specific important covariates for high-dimensional heterogeneous data [2]. proposed cumulative divergence (CD) metric and developed a model-free CD-based forward screening procedure. In [3], a distributed screening framework was utilized, which applies a correlation measure as a function of several component parameters and each of those components can be distributively estimated. With the components estimates, a final correlation estimate can be adopted for screening features [4]. proposed a model-free and data-adaptive feature screening method which is based on the projection correlation between two random vectors for ultra-high dimensional data. This approach is applicable for heavy tail and multivariate responses.

A large number of variable selection approaches based on regularization have been developed to tackle the high-dimensionality issue. One of the most popular and renowned regularization method, the Least Absolute Shrinkage and Selection Operator (LASSO) method, was proposed by Tibshirani [5]. The LASSO uses the $l_{1}$ penalty and minimizes the squared error. The major advantage of LASSO method is that it performs the variable selection and parameter estimation simultaneously. Unlike the ridge regression, the LASSO is able to shrink the coefficient estimate towards zero. Despite the popularity of the LASSO, many alternative choices of penalty functions are also available. Fan and Li [6] proposed the smoothly clipped absolute deviations (SCAD) penalty, which is a nonconvex penalty. Another example is the Dantzig selector (DS) method proposed by [7], which minimizes the maximum component of the gradient of the squared error function [6]. reviewed and summarized a family of well-established work on variable selection problems by using a penalized likelihood approach in the finite parameter settings and established the oracle properties for non-concave penalized likelihood estimators.

It was argued that the regularization methods cited above may not perform as expected due to the simultaneous challenges of computational expediency, statistical accuracy, and algorithm stability [8]. Thus, a large number of two-stage approaches have been proposed to improve the performance of the regularization methods and reduce the computational cost. In the first stage of these two-stage methods, the dimension of the data was reduced. One can choose from different dimension reduction methods to reduce the number of variables from very large to moderate. Then in the second stage, classic variable selection algorithms can be applied without the curse of high-dimensionality to identify the important features selected from the first stage. The choice of variable selection algorithms ranges from regularization to model selection criteria. Ideally, all of the important features are selected and only a few nuisance variables are kept in the first stage. Therefore, the first stage is usually referred to as the feature screening stage and the second stage as the post-screening stage.

The two-stage approach can be applied to linear models. Fan and Lv [9] proposed the sure independence screening (SIS) method to select important variables based on marginal Pearson correlation coefficient between each predictors and response variable in the first stage. By applying SIS in the first stage, we can select the features that have the strongest correlation with the response variable and reduce high-dimensionality to a relative moderate size. Following the first stage, appropriate regularization methods such as LASSO, SCAD, and Dantzig can be applied in the second stage to further select the important features. Those methods are referred as SIS-LASSO, SIS-SCAD, and SIS-DS.

To broaden the application of two-stage feature screening and variable selection, generalized linear models are involved, and they are popularized via McCullagh and Nelder [10]. In such models, a link function (often nonlinear) connects the mean of a response variable and linear combinations of predictors. A generalized linear model serves as a flexible and more general framework that can be used to build many types of regression models. The response variable is assumed to follow an exponential family distribution and does not have to be a normal distribution. With the release of normality assumption, generalized linear models can therefore be applied to a wide spectrum of data for modeling analysis. As an extension of the linear regression, generalized linear models are substantially utilized in a variety of fields, such as biomedical and educational research, social sciences, agriculture, environmental health, financial analysis, etc.

Sure independent screening method was demonstrated to be capable of efficiently selecting important predictors with low computational cost in linear models. Therefore, it is a natural extension to apply the feature screening method to generalized linear models. Fan and Song [11] extended the feature screening procedure for generalized linear models by ranking the marginal maximum marginal likelihood estimator (MMLE). This method ranks marginal regression coefficient of generalized linear model to screen the important features. It is able to dramatically reduce the dimension of the data and make the computation more feasible after the screening. Actually, the MMLE ranking is the same as the marginal correlation ranking in the linear model setting. Further, it does not depend on normality assumption and can be applied to other models. A variety of marginal screening procedures have been proposed by applied different types of correlations and for different types of models.

Even though some feature screening procedures such as MMLE and Kolmogorov filter [12] have been proposed for generalized linear models, those methods have their own limitations. MMLE approaches can select important predictors efficiently, but the computational cost for this method is relative high since it requires fitting the marginal model for each predictor. The Kolmogorov filter method is computationally fast, but the selection accuracy is relatively low compared with certain methods. Inspired by those two-stage feature screening approaches, we propose a two-stage feature screening approach for high-dimensional variable selection in generalized linear model with binary response variable. The point-biserial correlation [13] is a well-known correlation that can be used to measure the strength and the direction between one continuous variable and one binary variable. In the first stage, we can apply point-biserial correlation as a marginal index to check the correlation between each predictor and the response to reduce the dimension of the data to a moderate size. Then, we apply a regularization method to further select important predictors and build the final sparse model.

The primary objective of this paper is to develop a two-stage feature screening method called point-biserial sure independence screening (PB-SIS) for high-dimensional generalized linear models, aiming for high selection accuracy and low computational cost. The latter property is quite important in the era of big data, where the size of data sets becomes larger and never stops growing with the advancement of modern science and technology. We demonstrate that PB-SIS is a feature screening method with high efficiency.

Section 2 introduces generalized linear models. Section 3 presents the PB-SIS method and the two-stage point-biserial correlation screening procedure. Section 4 conducts a set of simulation studies to compare the performance of the proposed method with MMLE [11] and Kolmogorov filter method [12]. The predictors are set to have different strengths of pair-wise correlation and the response variable is generated by using different link functions. These simulations confirm the sure independence property and the accuracy and efficiency of PB-SIS. We demonstrate the effectiveness of PB-SIS with the application to one real data example in Section 5. Section 6 concludes and discusses.

## 2. Generalized Linear Models (GLMs)

Even though the sure independence screening (SIS) method proposed by Fan and Lv [9] provides a very useful and powerful tool for high-dimensional data analysis, it focuses on the linear models setting and its properties dependent on the joint normality assumptions. Fan and Song [11] also proposed a more general version of sure independence screening method for generalized linear models (GLMs), which ranks the maximum marginal likelihood estimator (MMLE) or maximum marginal likelihood itself. Assume that the response *Y* is from an exponential family with the canonical form:$$f_{Y}\left(y,\theta \right)=exp\left\{y\theta -b\left(\theta \right)+c\left(y\right)\right\},$$
where let $X=(X_{1},X_{2},\ldots ,X_{p})$ be the *p*-dimensional explanatory variables shown as the n×p design matrix. Denote $X_{ij}$ as the *i*th observation of the *j*th variable, then we have $X_{i}={\left(X_{i1},X_{i2},\ldots ,X_{ip}\right)}^{T}$. The b(·), c(·) are some unknown functions, and natural parameter θ. Then we have the following generalized linear model:$$E\left(Y|x\right)=b^{\prime }\left(\theta \left(x\right)\right)=g^{-1}\left(\beta_{0}+x^{T}\beta \right),$$
where g(·) is the link function, $\beta_{0}$ is an unknown scalar. Let $\beta ={\left(\beta_{1},\beta_{2},\ldots ,\beta_{p}\right)}^{T}$ be a *p*-dimensional unknown vector. Let $\left\{x_{i},Y_{i}\right\}$, i=1,2,…,n, be an independent and identically distributed sample from a population {x,Y}. For the MMLE method, ${\hat{\beta }}_{j}^{M}$ for the *j*th predictor $X_{j}$ is defined as
$${\hat{\beta }}_{j}^{M}={\left({\hat{\beta }}_{j0}^{M},{\hat{\beta }}_{j1}^{M}\right)}^{T}=\underset{\beta_{j0},\beta_{j1}}{argmin}\frac{1}{n}\sum_{i=1}^{n}\ell \left(Y_{i},\beta_{j,0}+\beta_{j1}X_{ij}\right),$$
where ℓ(y,θ)=−yθ(x)+b(θ)−c(y) is the log likelihood function. Ref. [11] considered to rank magnitude of the marginal regression coefficients ${\hat{\beta }}_{j1}^{M}$ to select important features and defined the selected submodel as
$${\hat{M}}_{\gamma_{n}}=\{i\leq j\leq p:|{\hat{\beta }}_{j}^{M}\left|>\gamma_{n}\right\},$$
where $\gamma_{n}$ is a pre-specified threshold. The dimension of *p* will dramatically decrease to a moderate size when we choose a large value of $\gamma_{n}$.

To establish the theoretical properties of MMLE, Fan and Song [11] defined the population version of the marginal likelihood maximize as
$$\beta_{j}^{M}={\left(\beta_{j0}^{M},\beta_{j1}^{M}\right)}^{T}=\underset{\beta_{j0},\beta_{j1}}{}argminE\left[\ell \left(Y_{i},\beta_{j,0}+\beta_{j1}X_{ij}\right)\right],$$
where *E* denotes the expectation under the true model. Based on this population aspect, it was shown that the marginal regression parameter $\beta_{j1}^{M}=0$ if and only if $cov(Y,X_{j})=0$, for j=1,2,…,p. Thus, $\beta_{j1}^{M}\neq 0$ when the important features are correlated with the response variable. Define the true model as $M_{★}=\left\{1\leq j\leq p:\beta_{j}\neq 0\right\}$ with the size s=|M|. Under some conditions, if $|cov\left(Y,X_{j}\right)|\geq c_{1}n^{-\kappa }$ for j∈M and some $c_{1}>0$, then we have
$$\underset{j\in M_{★}}{min}\left|\beta_{j1}^{M}\right|\geq c_{2}n^{\kappa },$$
for some $c_{2}$, κ>0. Thus, the marginal signals $\beta_{j1}^{M}$’s are stronger than the stochastic noise provided that $X_{j}$’s are marginally correlated with *Y*.

Fan and Song [11] also showed that under proper regularity conditions, this procedure has sure screening property and size control property if $\gamma_{n}$ follows an ideal rate. Under certain conditions, we have
$$Pr(M_{★}\subset M_{\gamma_{n}})\to 1\;as\;n\;\to \infty .$$
where $\gamma_{n}$ = $cn^{1-2k}$ for some 0<k<1/2 and c>0. The $M_{*}=\left\{1\leq j\leq p:\beta_{j}^{*}\neq 0\right\}$ is the true index set of model.

## 3. Feature Screening Methodology for Generalized Linear Models via Point-Biserial Correlation

We propose a two stage feature screening method for GLMs variable selection by using point-biserial correlation. In the first stage, we use point-biserial correlation as a marginal utility to rank predictors and select the submodel by using some predefined threshold. This step can reduce the number of features from a very large scale to a moderate size in a computationally fast manner. Then in the second stage, we apply a regularization method, such as LASSO, SCAD or MCP, to further shrink the number of parameters and find the final sparse model from the screened set we got from the first stage. This proposed method is referred as the two-stage PB-SIS.

We remark that [9] demonstrated that the two-stage methods which combine independence screening and penalized method outperform an one-step penalized method. The effectiveness of the two-stage method is guaranteed by the sure screening property. The sure screening properties mean all important predictors are selected in the reduced model almost surely, e.g., the sure screening property for PB-SIS guarantees that PB-SIS is able to retain all of the variables from the true model in the screened submodel with probability going to one as the sample size goes to infinity, and the convergence rate is exponential. It can be shown that PB-SIS possesses the sure independence property under certain regularity conditions and that the PB-SIS method can select all of the important variables in the model with probability one.

### 3.1. Point-Biserial Correlation and Its Asymptotic Distribution

Let *Y* be a binary variable with two classes $y_{0}=0,y_{1}=1$, and again $X=(X_{1},X_{2},\ldots ,$$X_{p}{)}^{T}$ be a n×p covariate matrix. Given *n* independent identically distribution random sample $X_{i}={\left(X_{i1},\ldots ,X_{ip}\right)}^{T}$. Let $X_{ij}$, i=1,2,…,n, j=1,…,p, be the *i*th sample of the *j*th covariate. To investigate the point-biserial correlation between *Y* and $X_{j}$, j=1,2,…,p, we consider the correlation between each $X_{j}$ and *Y*.

For each *j*, consider $\left(X_{i},Y_{i}\right)$, i=1,2,…,n, a sequence of independent random vectors. Assume $Y_{i}$ have the Bernoulli distribution:(1)$$P\left(Y_{i}=1\right)=p_{1},\;P\left(Y_{i}=0\right)=p_{0},$$
where 0<p<1 and $p_{1}+p_{1}=1$. Assume $X_{i}$ have the mixture normal distribution which can be written as either the distribution function *F* or the density function *f*:(2)$$F\left(x\right)=p_{1}F_{1}\left(x\right)+p_{0}F_{0}\left(x\right)\;\;\mathrm{and}$$
$$f\left(x\right)=p_{1}f_{1}\left(x\right)+p_{0}f_{0}\left(x\right),$$
where
$$F_{k}=P\left(X\leq x|X=k\right)=\int_{-\infty }^{x}\frac{1}{\sigma \sqrt{2\pi }}e^{-\frac{{\left(z-\mu_{k}\right)}^{2}}{2\sigma^{2}}}dz,\;k=0,1.$$
The random variable *Z* is asymptotically normal with a mean of μ and a variance of $\sigma^{2}$.

Consider *X* normally distributed in $Z_{0}$ and $Z_{1}$ separately with different mean $\mu_{1}$, $\mu_{0}$ and same variance $\sigma_{1}^{2}$, $\sigma_{0}^{2}$, where we have
$$\mu_{1}=E\left(X|Y_{i}=1\right),\;\mu_{0}=E\left(X|Y_{i}=0\right),$$
$$\sigma_{1}^{2}=Var\left(X|Y_{i}=1\right),\;\sigma_{0}^{2}=Var\left(X|Y_{i}=0\right),\;\mathrm{and}\;\sigma_{1}^{2}=\sigma_{0}^{2}=\sigma .$$
Thus, the point-biserial correlation can be defined as
$$r_{pb}=\frac{\sum_{i=1}^{n}\left(X_{i}Y_{i}-n\bar{X}\bar{Y}\right)}{\sqrt{\sum_{i=1}^{n}{\left(X_{i}-\bar{X}\right)}^{2}}\sqrt{\sum_{i=1}^{n}{\left(Y_{i}-\bar{Y}\right)}^{2}}}.$$

Since $Y_{i}$ has the Bernoulli distribution with probability in Equation (Equation 1), the mean and variance of random variable *Y* are
$$E\left(Y\right)=1\left(p_{1}\right)+0\left(p_{0}\right)=p_{1}\;\;\mathrm{and}$$
$$Var\left(Y\right)={\left(1-p_{1}\right)}^{2}\left(p_{1}\right)+{\left(0-p_{0}\right)}^{2}\left(p_{0}\right)=p_{1}p_{0}.$$
Since *X* follows the mixture normal with CDF in Equation (Equation 2), the expected value and variance of *X* are
$$\begin{array}{cc} E(X) & =p_{1}\mu_{1}+p_{0}\mu_{0}\;\;\mathrm{and}\\ Var(X) & =\sigma^{2}\left(1+p_{0}p_{1}\frac{{\left(\mu_{1}-\mu_{0}\right)}^{2}}{\sigma^{2}}\right). \end{array}$$

Denote the standardized difference of means $\mu_{1}$ and $\mu_{0}$, $\frac{\mu_{1}-\mu_{0}}{\sigma }$, as Δ. Thus, the variance of random variable *X* can be written as
$$Var\left(X\right)=\sigma^{2}\left(1+p_{0}p_{1}\Delta^{2}\right).$$
Then, we can derive the expected value of product of *X* and *Y*. Since the product of XY is zero when X=0 or Y=0, the expected value of XY only takes the value when Y=1. Therefore, we have
$$E\left(XY\right)=p\mu_{1}.$$
Now we can find the population correlation coefficient *X* and *Y*
$$\begin{array}{cc} \rho (X,Y) & =\frac{Cov(X,Y)}{\sigma_{x}\sigma_{Y}}\\ \; & =\frac{\mu_{1}-\mu_{0}}{\sigma }\sqrt{\frac{p_{1}p_{0}}{1+p_{1}p_{0}\Delta^{2}}}, \end{array}$$
which has the form $\rho \left(X,Y\right)=\Delta \sqrt{\frac{p_{1}p_{0}}{1+p_{1}p_{0}\Delta^{2}}}$ and it has a natural estimator, $r_{pb}$.

Remark 1 states the asymptotic distribution of point-biserial correlation which can be easily extended from [13].

**Remark** **1.**
*Let random variable Y have a Bernoulli distribution and random variable X have mixture normal distribution with CDF in form (Equation 2), then the point-biserial correlation, $r_{pb}$, between X and Y has the asymptotic distribution*

$$r_{pb}∼N\left[\rho ,\frac{4p_{1}p_{0}-\rho^{2}\left(6p_{1}p_{0}-1\right)}{4np_{1}p_{0}}{\left(1-\rho^{2}\right)}^{2}\right].$$



### 3.2. Two-Stage Point-Biserial Correlation Screening Procedure

We consider using the point-biserial correlation to measure the correlation between $X_{j}$, j=1,2,…,p, and *Y*. We define the following index
$$\omega_{j}=\frac{E[(X_{j}-E\left(X_{j}\right)\left(Y-E\left(Y\right)\right]}{\sqrt{Var(X_{j})}\sqrt{Var(Y)}},$$
as a marginal utility measure for screening. Intuitively, we can see that if $X_{j}$ and *Y* are independent or close to independent, then $\omega_{j}=0$ or $\omega_{j}$ is very close to 0. On the other hand, if $X_{j}$ and *Y* have strong correlation, $\omega_{j}$ is close to −1 or 1. Thus, we can rank the marginal $\omega_{j}$ value to select important features that have higher correlation with the response variable.

A natural estimator for $\omega_{j}$ can be defined as
$${\hat{\omega }}_{j}=\frac{\sum_{i=1}^{n}\left(X_{ij}Y_{i}\right)-n\bar{X_{j}}\bar{Y}}{\sqrt{\sum_{i=1}^{n}{\left(X_{ij}-\bar{X_{j}}\right)}^{2}}\sqrt{\sum_{i=1}^{n}{\left(Y_{i}-\bar{Y}\right)}^{2}}}.$$

Based on ${\hat{\omega }}_{j}$, we propose a two-stage screening procedure for high-dimensional GLMs with binary response variable. In the first stage, we compute sample point-biserial correlation ${\hat{\omega }}_{j},j=1,2,\ldots ,p$ for each predictor. Then we can sort the magnitudes of all the components of $\hat{\omega }={\left({\hat{\omega }}_{1},{\hat{\omega }}_{2},\ldots ,{\hat{\omega }}_{p}\right)}^{T}$ in a decreasing order and select a submodel as
(3)$${\hat{M}}_{d}=\{j:1\leq j\leq p:|{\hat{\omega }}_{j}\left|\mathrm{is}\;\mathrm{among}\;\;\mathrm{the}\;\mathrm{first}\;d\;\mathrm{largest}\;\mathrm{of}\;\mathrm{all}\right\},$$
where the submodel size *d* is smaller than the sample size *n*. Thus, we can reduce the high dimension *p* to the moderate size *d*. As Ref. [9] suggested, the submodel size *d* could be set as ⌊n/log(n)⌋, where the ⌊a⌋ refers as the floor function of *a*. The submodel (Equation 3) has the equivalent from
$${\hat{M}}_{d}=\{1\leq j\leq p:|{\hat{\omega }}_{j}\left|>\gamma \right\},$$
where the *d* or γ is a predefined threshold value. This proposed procedure is referred to as point-biserial correlation sure independence screening (PB-SIS).

Although the PB-SIS method can reduce the high dimensionality *p* to a moderate size *d*, we can apply a penalized method in the second stage to further select important variables to find the final sparse model. In the second stage, a penalty regression procedure, such as the least absolute shrinkage and selector operator (LASSO), can be applied to further select important variables and estimate the coefficients in model. LASSO is a shrinkage method which places a constraint on the absolute values of the parameter in a model. It is the most popular approach for selecting significant variable and estimating coefficients simultaneously. The LASSO estimates is defined as
(4)$${\hat{\beta }}^{lasso}=\underset{\left(\beta_{0},\beta \right)\in R^{d+1}}{argmin}\left\{\frac{1}{2}{\left(y_{i}-\beta_{0}-x_{i}^{T}\beta \right)}^{2}+\lambda \sum_{j=1}^{d}\left|\beta_{j}\right|\right\}.$$
Ref. [14] proposed fast regularization path for GLMs via coordinate descent. This method can handle LASSO penalty for estimation problems efficiently.

For solving Equation (Equation 4) in generalized linear models setting, Ref. [14] considered a coordinate descent steps. Suppose we have estimates ${\tilde{\beta }}_{0}$ and ${\tilde{\beta }}_{l}$ for l≠j, and we would like to partially optimize with respect to $\beta_{j}$. Let $R(\beta_{0},\beta )$ be the objective function in Equation (Equation 4). The gradient at $\beta_{j}=\tilde{\beta_{j}}$ could be computed if ${\tilde{\beta }}_{j}\neq 0$. Thus, if ${\tilde{\beta }}_{j}>0$, then we have
$$\frac{\partial R}{\partial \beta_{j}}{|}_{\beta =\tilde{\beta }}=-\frac{1}{N}\sum_{i=1}^{N}x_{ij}\left(y_{i}-{\tilde{\beta }}_{0}-x_{i}^{T}\tilde{\beta }\right)+\lambda .$$

Ref. [15] showed that after a simple calculation, the coordinate-wise update has the form:$${\tilde{\beta }}_{j}\leftarrow S\left(\frac{1}{N}\sum_{i=1}^{N}x_{ij}\left(y_{i}-{\tilde{y}}_{i}^{\left(j\right)}\right),\lambda \right)$$
where ${\tilde{y}}_{i}^{j}={\tilde{\beta }}_{0}+\sum_{l\neq j}x_{il}{\tilde{\beta }}_{l}$ is the fitted value excluding the contribution from $x_{ij}$, and $y_{i}-{\tilde{y}}_{i}^{\left(j\right)}$ is the partial residual for fitting $\beta_{j}$. The S(z,γ) is the soft-threshold operator with the value: $$sign\left(z\right){\left(|z|-\gamma \right)}_{+}=\{\begin{array}{cc} z-\gamma & if\;z>0\;and\;\gamma <|z|\\ z+\gamma & if\;z<0\;and\;\gamma <|z|\\ 0 & if\;\gamma \geq |z|. \end{array}$$
The details of this derivation are showed in [16].

Since we focus on feature screening for GLMs with binary response question, the logistic regression model is commonly used. We would like to investigate the model optimization and estimation for penalized logistic regression as follow. As we discussed before, the logistic regression model can be represented by the class-conditional probabilities through a linear function of the predictors as
(5)$$P\left(G=1|x\right)=\frac{1}{1+e^{-(\beta_{0}+x^{T}\beta )}},$$
$$P\left(G=0|x\right)=\frac{1}{1+e^{+(\beta_{0}+x^{T}\beta )}},$$
where P(G=1|x)=1−P(G=0|x). This can imply the logistic regression formula:$$\log\frac{P(G=1|x)}{P(G=0|x)}=\beta_{0}+x^{T}\beta .$$
Let $p\left(x_{i}\right)=P\left(G=1|x_{i}\right)$ be the probability in Equation (Equation 5) for observation *i* at a particular value for the parameters $\left(\beta_{0},\beta \right)$, then Ref. [14] maximized the penalized log-likelihood:(6)$$\underset{\left(\beta_{0},\beta \right)\in R^{d+1}}{min}\left[\frac{1}{N}\sum_{i=1}N\left\{I\left(g_{i}=1\right)\log p\left(x_{i}\right)+I\left(g_{i}=0\right)\log\left(1-p\left(x_{i}\right)\right)\right\}-\lambda P_{\lambda }\left(\beta \right)\right].$$
Denote $y_{i}=I\left(g_{i}=1\right)$, then the penalized log-likelihood in Equation (Equation 6) can be represented
(7)$$\ell \left(\beta_{0},\beta \right)=\frac{1}{N}\sum_{i=1}^{N}y_{i}\left(\beta_{0}+x_{i}^{T}\beta \right)-\log\left(1+e^{\left(\beta_{0}+x_{i}^{T}\beta \right)}\right),$$
which is a concave function of the parameter. For the unpenalized log-likelihood problem, we could apply Newton’s method to work on maximizing iteratively reweighted least squares. We could form a quadratic approximation (Taylor expansion) for the log-likelihood to estimate $\left({\tilde{\beta }}_{0},\tilde{\beta }\right)$ as
(8)$$\ell_{Q}\left(\beta_{0},\beta \right)=-\frac{1}{2N}w_{i}{\left(z_{i}-\beta_{0}-x_{i}^{T}\beta \right)}^{2}+C\left({\tilde{\beta }}_{0},\tilde{\beta }\right),$$
where
(9)$$\begin{array}{cc} \; & z_{i}={\tilde{\beta }}_{0}+x_{i}^{T}\tilde{\beta }+\frac{y_{i}-\tilde{p}\left(x_{i}\right)}{\tilde{p}\left(x_{i}\right)\left(1-\tilde{p}\left(x_{i}\right)\right)}\;\;\mathrm{and}\\ \; & \omega_{i}=\tilde{p}\left(x_{i}\right)\left(1-\tilde{p}\left(x_{i}\right)\right), \end{array}$$
and $\tilde{p}\left(x_{i}\right)$ is evaluated at current parameter. The last term in Equation (Equation 8) is constant, and $z_{i}$ is the working response and $\omega_{i}$ is weights in Equations (Equation 9). The Newton update could be obtained by minimizing *ℓ* in Equation (Equation 8). Ref. [14] proposed the coordinate descent approach to optimize the penalized log-likelihood in (Equation 7), which is similar as the Newton’s method. As they suggested, we can create an outer loop which computes the quadratic approximation $\ell_{Q}$ about the current parameters $\left({\tilde{\beta }}_{0},\tilde{\beta }\right)$ for each value of λ. Then use coordinate descent to solve the penalized weighted least-squares problem as
(10)$$\underset{\left(\beta_{0},\beta \right)\in R^{d+1}}{min}\left\{-\ell_{Q}\left(\beta_{0},\beta \right)+\lambda P_{\alpha }\left(\beta \right)\right\}.$$
To implement this algorithm, we need to use a sequence of loops at the same time. We can use the outer loop to decrement λ, use the middle loop to update the quadratic approximation $\ell_{Q}$ using the current parameter $\left({\tilde{\beta }}_{0},\tilde{\beta }\right)$, and apply the inner loop to run the coordinate descent algorithm on the penalized weighted least squares problems in objective function (Equation 10). We then iterate those nested loops until convergence.

Besides LASSO penalty, the smoothly clipped absolute deviation (SCAD) penalty [6] and the minimax concave penalty (MCP) [17] also can be applied in the second stage to further select important predictors and estimate the coefficients. The SCAD and MCP are concave penalties that satisfy the oracle properties. It means that those two penalized methods can correctly select important variables and estimate coefficients with high probabilities if certain regularity conditions are met. For the SCAD penalty, Ref. [6] proposed a local quadratic approximation (LQA) algorithm to find the optimal solutions. However, once a coefficient is set to zero at any iteration, it will keep staying at zero and the corresponding variable is removed from the final model for LQA algorithm. Ref. [18] proposed the majorization-minimization (MM) approach to optimize a perturbed version of LQA by bounding the denominator away from zero. Besides, Ref. [19] proposed a local linear approximation (LLA) algorithm to approximate the concave penalized solution by repeatedly using the algorithms for the LASSO penalty. However, most of those optimization methods are for linear models. Ref. [20] proposed a majorization minimization by coordinate descent (MMCD) to find the optimal solutions of a concave penalized in GLMs, with emphasis on the logistic regression. They implemented this algorithm for a penalized logistic regression model using the SCAD and MCP penalties.

Since this algorithm can not run λ all the way to zero if *p* is much greater than *n* since the saturated logistic regression fit is undefined, it is necessary to apply the first stage of our proposed method first to reduce the number of parameters to a moderate size. Then we use a penalized method, such as LASSO, SCAD and MCP, at the second stage to obtain the final model. This algorithm is easily to implement by using R package SIS. By applying the SIS, one can use cross-validation (CV), AIC [21], BIC [22] or EBIC [23] to choose tuning parameter λ.

The summary of two-stage PB-SIS method is provided in Algorithm 1.
**Algorithm 1** Two-stage PB - SIS Algorithm.1:Compute the point-biserial correlation between $x_{j}$ and *y* as ${\hat{\omega }}_{j}$ and rank the magnitude of the absolute value of marginal correlation ${\hat{\omega }}_{j}$.2:Choose the predefined threshold value *d* and take the selected submodel to be ${\hat{M}}_{d}=\{j:1\leq j\leq p:|{\hat{\omega }}_{j}|$ is among the first *d* largest of all}, where *d* is some predefined threshold.3:Start with all variables in the submodel ${\hat{M}}_{d}$, then apply a penalized method, such as LASSO, SCAD or MCD, to further select important variables and estimate coefficients $\left({\tilde{\beta }}_{0},\tilde{\beta }\right)$.

## 4. Simulations

We will conduct Monte Carlo simulations to evaluate the performance for the proposed PB-SIS method with some existing feature screening methods for generalized linear models (GLMs), like sure screening by ranking the magnitude likelihood estimator (MMLE) [11], and screening for binary classification based on the Kolmogorov-Smirnov statistic (Kolmogorov Filter) [12]. We will also check the performance of two-stage PB-SIS method with different penalized methods by using different tuning parameter selection criteria.

### 4.1. Simulation Settings

In each example, the data $\left(X_{1}^{T},Y_{1}\right),\left(X_{2}^{T},Y_{2}\right),\ldots ,\left(X_{n}^{T},Y_{n}\right)$ are independent copies of a pair $\left(X^{T},Y\right)$, where the conditional distribution of the response *Y* given X=x is a binomial distribution with probability of success $\pi_{i}$. We generate $x={\left(X_{1},X_{2},\ldots ,X_{p}\right)}^{T}$ from multivariate normal distribution with mean 0 and covariance matrix $\Sigma ={\left(\sigma_{ij}\right)}_{p\times p}=\rho^{\left|i-j\right|}$. We set up 5 different ρ values from small to large to generate *X* with different correlation strength among the *p* predictors. There are independence (ρ=0), low correlation (ρ=0.2), moderate correlation (ρ=0.4), high correlation (ρ=0.6) and very high correlation (ρ=0.8). We vary the size of the non-sparse set of coefficients as s=2,3,4 with vary signals and set up the number of parameter with p=200 and p=600. Besides, we apply one link function, logit, to generate the binomial proportion $\pi_{i}$, then generate the binary response variable *Y*. For each link function, we consider 6 different models which are presented in Table 1 with different covariates. The true coefficients for these 6 models are β=(2,3), β=(2,−3), β=(2,3,3), β=(2,−3,3), β=(2,3,3,3), and β=(2,−3,3,−3) and the same constant term $\beta_{0}=1$. Note that these parameters are randomly selected and some easily recognizable numbers are chosen for brevity. The patterns and trends of the simulation results do not depend on the parameter values. Thus, the proposed PB-SIS method is compared with MMLE and Kolmogorov filter method under all 2×6=18 simulation settings. All simulation results are based on 1000 replicates.

For each simulation, we use the proportion of submodels $M_{d}$ with size *d* that contain all the true predictors among 1000 replications, $P_{1}$, and computing time to evaluate the performance for each setting. For the threshold value *d*, we follows [9] and choose *d* to be $d_{1}=\left\lfloor n/\log n\right\rfloor$, $d_{2}=2\left\lfloor n/\log n\right\rfloor$ and $d_{3}=3\left\lfloor n/\log n\right\rfloor$ throughout our simulations to empirically examine the effect of the cutoff, where the ⌊n/logn⌋ means the floor function of n/log(n). Since in our simulation setting, we take n=100, we have $d_{1}=21$, $d_{2}=43$, and $d_{3}=65$. We also evaluate each method by summarizing the median minimum model size (MMMS) of each selected models and its robust estimate of the standard deviation (RSD). RSD is the interquantile range (IQR) divided by 1.34, which is given by [11].

For the principle to define the value of *d*, Ref. [9] set d=n/log(n) as one way of choices for *d*, and this way is conservative yet effective. Their preference is to select sufficiently many features in the first stage, and when *d* is not very small, the selection results are not very sensitive to the choice of *d*. It is obvious that larger *d* means larger probability of including the true model $M_{*}$ in the submodel $M_{d}$. Provide that d=n/log(n) is large enough, we can use it as the threshold. Doing so can detect all significant predictors in the selected subset and the $P_{1}$ value is large. Therefore, the principle for choosing *d* is to obtain a relatively large value of *d* to ensure the selection of the first stage can include all important predictors in the submodel $M_{d}$. The simulation results in the next subsection will show that taking $d_{1}=\left\lfloor n/\log n\right\rfloor$, $d_{2}=2\left\lfloor n/\log n\right\rfloor$ and $d_{3}=3\left\lfloor n/\log n\right\rfloor$ as thresholds results in the $P_{1}$ values being close to 1, verifying that these thresholds perform effectively in the proposed feature screening method.

### 4.2. Presentation of Simulation Results for Logit Models

We present a series of simulation results where the response variable is generated from GLMs for binary data by using logit link. For the link function, we will summarize simulation results for 6 different models in Table 1. The proportion $P_{1}$ and computing time are tabulated in first 6 tables and the MMMS and the associated RSD are summarized in Tables 7–12 for each link.

The simulation results for model 1 to model 6 where data is generated from logit link are tabulated in Table 2, Table 3, Table 4, Table 5, Table 6 and Table 7. From Table 2, we can see that the all proportions $P_{1}$ are close to 1, which illustrates the sure screening property. MMLE screening procedure usually has highest proportion $P_{1}$ than the other two methods, but it takes much longer computing time than PB-SIS method and Kolmogorov-filter method in all settings. Even through the proportion $P_{1}$ for PB-SIS is slightly lower than MMLE when ρ=0 and ρ=0.2, the difference is very small. The biggest difference for proportion $P_{1}$ is only 1.3% between PB-SIS and MMLE when ρ=0 and p=600. When ρ is greater than 0.4, the PB-SIS and MMLE have the exact same proportion $P_{1}$. But when we consider about computational cost, the PB-SIS method can be implemented much fast than the MMLE method. The average computing time for the PB-SIS and MMLE methods in logit model 1 are 41.85 seconds and 579.18 seconds when p=200, and 282.05 seconds and 1289.69 seconds when p=600. The computing time for MMLE is almost 6.74 times and 2.23 times longer than the PB-SIS method when p=200 and p=600. The Kolmogorov filter method has lowest proportion $P_{1}$ and moderate computing time in each setting. Since we assign all coefficients are positive in logit model 1, the proportions $P_{1}$ do not dependent on the independence assumption. Even for the highly correlated predictors, all three feature screening methods still can successfully select all the true predictors. For example, the proportions $P_{1}$ are all equals to 100% when ρ=0.6 and ρ=0.8. Besides, the proportion $P_{1}$ decreases as the dimensionality increases. As the number of features increases from p=200 to p=600, the proportions $P_{1}$ decrease in most settings.

The proportion $P_{1}$ and computing time for logit model 2 are reported in Table 3. In logit model 2, the two true covariates are assigned different signs. All $P_{1}$ of PB-SIS and MMLE are still very close. It means those two screening procedures perform equally well in most of settings. However, when we compare the computing time for the different methods, we can observe that PB-SIS takes much shorter computing time than MMLE in all settings. If we compare covariance structures with different ρ’s, those predictors are independent to each other (ρ = 0) and predictors have low correlation (ρ = 0.2) settings typically perform better than those with high (ρ = 0.6) or very high ρ=0.8 correlation settings for all three screening procedure. This is due to the probabilities of selecting some unimportant variables are inflated by the adjacent important ones when the predictors are highly correlated. Then some unimportant predictors may be selected since those predictors have strong correlation with the true predictors and it weakens the probabilities of selecting all true predictors.

Table 4 depicts the proportion $P_{1}$ and computing time for logit model 3. Similar conclusions can be drawn from Table 4 as from Table 2. All proportions $P_{1}$ of all three screening approaches are close to one. It means those three approaches are able to select all important predictors in this setting. As the submodel size *d* increases, the proportions $P_{1}$ for all three approaches increase as well. Thus increasing the submodel size *d* is helpful for increasing the proportion $P_{1}$. The computing time does not change too much as the submodel size *d* increases. If we would like to get higher proportion $P_{1}$, we can choose a larger threshold *d*. However, the larger threshold *d* means the model will become more complex. There is a trade off between the model complexity and the selection accuracy. Our suggestion is to choose the smaller submodel model size d=⌊n/log(n)⌋, since the small growth of the proportion $P_{1}$ is not worth the increasing of twice or three times of model complexity.

Table 5 reports the proportion $P_{1}$ and computing time for logit model 4. In logit model 4, the three true covariates are assigned different signs. The PB-SIS and MMLE perform equally well and PB-SIS approach is more efficient when ρ=0, ρ=0.2, ρ=0.4 or ρ=0.6. However, when predictors are highly correlated (ρ=0.8), all three feature screening fail to detect important predictors. This is because when predictors are highly correlated (ρ = 0.8), each predictor’s contribution to the response variable is cancelled out, especially for the predictors have opposite sign.

The proportion $P_{1}$ and computing time for logit model 5 and logit model 6 are summarized in Table 6 and Table 7. For logit model 5, we observe a qualitative pattern similar to logit model 1 and logit model 3. The PB-SIS and MMLE approaches perform equally well, and the PB-SIS approach yields a comparable computing time. The Kolmogorov filter approach performs a little bit worse than the PB-SIS in both selection accuracy and computing time. We also observe that the proportion $P_{1}$ increases as the correlation ρ increases. From Table 7, the simulation results show the PB-SIS and MMLE perform equally well in selection accuracy, while the PB-SIS approach has lower computational cost than MMLE when predictors are independent or have lower correlation. Similar to logit model 1 and logit model 3 simulation results, when predictors are highly correlated, all three feature screening approaches tend to fail select important predictors.

Table 8 summarizes the MMMS which contains all true predictors for logit model 1 and its RSD. Those two values could be used to measure the effectiveness of a screening method. The MMMS value can avoid the issues of choosing different threshold *d*. From Table 8, we can observe that the PB-SIS and MMLE methods perform equally well and Kolmogorov filter approach performs a little bit worse than the PB-SIS and MMLE approaches in all settings. The Kolmogorov filter has a little bit larger RSD due to some outliers, which makes the minimum model size spread out in some cases. For the high correlation and very high correlation settings, the RSD values for PB-SIS and MMLE are larger, which means the minimum model size has higher variability when covariates are highly correlated to each other.

Table 9 depicts the MMMS and RSD for logit model 2. We can observe the similar results as logit model 1. The PB-SIS and MMLE still perform well in selecting all important variables when predictors are independent or have low correlation. However, all three feature screening procedures fail to detect important predictors when predictors are highly correlated (ρ = 0.8), especially for Kolmogorov filter method. For example, when the correlation is high, the MMMS of Kolmogorov filter are 16 and 33 for *p* = 200 and 600, and the RSD values even achieve 30.60 and 79.85 when p=200 and p=600. This means the minimum size models containing all important predictors are very spread out over the 1000 replications and may exist some outliers. This is mainly because each predictor’s contribution to the response variable is cancelled out when they are of the different signs and highly correlated to each other.

Table 10 summarizes the MMMS and RSD for logit model 3. The PB-SIS and MMLE approaches are more robust to select important predictors than Kolmogorov filter in most of settings. The MMMS value for PB-SIS and MMLE are almost same in all settings, and MMLE usually has smallest RSD values among all three feature screening procedures. The Kolmogorov filter method still performs a little bit worse than the PB-SIS and MMLE methods. In general, these three screening approaches do not make a big difference when the number of true predictors is small and of the same signs.

Table 11 presents the simulation results for logit model 4 in terms of MMMS and the associated RSD. The simulation results illustrate that the PB-SIS and MMLE have more effective and consistent performance than Kolmogorov filter method when ρ = 0, 0.2 or 0.4. In addition, we also notice that for the different dimension and correlation levels, the MMMS and the associated RSD usually increase as the dimension increases or the correlation level increases. When predictors are highly correlated, the PB-SIS, MMLE and Kolmogorov filter methods fail to select important predictors. For example, when ρ = 0.8, the MMMS of PB-SIS, MMLE and Kolmogorov filter procedures are 105, 105 and 144 for *p* = 200 and 140, 140 and 199 for *p* = 600, which are much larger than our true model size 3.

The simulation results for logit model 5 about the MMMS and the associated RSD are presented in Table 12. The overall pattern of logit model 5 is similar to logit model 1 and 3. The PB-SIS and MMLE methods still outperform Kolmogorov filter method in selection effectiveness. The Kolmogorov filter method has larger MMMS and the associated RSD than PB-SIS and MMLE in almost all settings.

The simulation results of MMMS with the associated RSD for logit model 6 are summarized in Table 13. From Table 13, we can observe that as the correlation increases, the MMMS and the associated RSD usually increase as well for all PB-SIS, MMLE and Kolmogorov filter approaches. In addition, we also see that as the dimension increases from 200 to 600, the MMMS also increases for all three feature screening approaches. Among the all approaches, the PB-SIS method usually can achieve smallest MMMS value in most settings. When predictors are highly correlated, all three feature screening methods fail to select important predictors. As we discussed before, this is due to the contribution of predictors with opposite signs may cancel out when predictors are highly correlated.

### 4.3. Simulations in Two-Stage Approach

We investigate the selection performance of two-stage PB-SIS method with different penalties. We consider the LASSO penalty, SCAD penalty and MCP along with four tuning parameter selection criteria: cross-validation(CV), Akaike information criterion (AIC), Bayesian information criterion (BIC) and Extended Bayesian information criteria (EBIC). In this section, only the logit link is applied to generate the binomial proportion $\pi_{i}$ and the binary response *Y*. We use the same model settings as Section 4.1 and are presented in Table 1. In the first stage, PB-SIS is conducted to obtain the submodel $M_{d}$ with size d=⌊n/log(n)⌋. Then in the second stage, three different penalized methods are applied to further select important predictors and recover final sparse model. All the simulation results are based on 1000 replicates.

We evaluate the two-stage PB-SIS performance based on the $P_{2}$, the proportion of final models containing all the true predictors among 1000 iterations and the mean of the final model size. The proportion $P_{2}$ and mean model size are summarized for model 1 to model 6 in Table 14, Table 15, Table 16, Table 17, Table 18 and Table 19 and the mean of the final model size after regularization is reported in the parentheses. We use package SIS in R to implement the penalized methods in the second stage. The tune.fit function in SIS package fits a generalized linear model via penalized maximum likelihood, with available penalties such as LASSO, SCAD and MPC as indicated in the glmnet and ncvreg packages. The number of folds used in cross-validation is 10 and loss function used in selecting the final model is deviance.

The proportion $P_{2}$ and mean model size for model 1 and model 2 are tabulated in Table 14 and Table 15. For model 1 and model 2, the number of true parameters are both two. In general, we can observe that the PB-SIS+LASSO two-stage approaches with different tuning selection criteria have the higher proportions $P_{2}$, while the PB-SIS+MCP two stage approaches with different tuning parameter selection criteria yield the sparsest models among all three different penalties. Even though the PB-SIS+LASSO two stage approaches usually have highest proportion $P_{2}$, they also give us the largest final models size for all different tuning parameter selection criteria. Furthermore, the PB-SIS+SCAD two-stage approach by using EBIC to select tuning parameter occasionally fails to select important predictors. For example, in Table 14, the proportions $P_{2}$ for the PB-SIS+SCAD two-stage approach by using EBIC to select tuning parameter are just 0.742 and 0.599 when p=200 and p=600, which are smallest among all two-stage approaches with different penalties. We also notice that as the dimension *p* increases, the proportion $P_{2}$ decreases and the mean model size increases for all three penalties.

Table 16 and Table 17 summarize the proportion $P_{2}$ and mean model size for model 3 and model 4. Model 3 and model 4 both contain three true parameters. For those two models, we observe similar overall pattern as model 1 and model 2. The final models which are selected by the PB-SIS+MCP two-stage approach with different tuning parameter selection criteria usually have smallest model size among the three penalties. The PB-SIS+SCAD two-stage approaches with different tuning parameter selection criteria return the moderate size final models and the PB-SIS+LASSO two-stage approaches with different tuning parameter selection criteria return the largest size final models. If we consider the proportion $P_{2}$, the PB-SIS+LASSO two-stage approaches with different tuning parameter selection criteria have the largest proportion $P_{2}$. We can conclude that the PB-SIS+LASSO two-stage approach performs better in selection accuracy and the PB-SIS+MCP two-stage approach performs better in finding the sparsest model.

The simulation results for model 5 and model 6 about proportion $P_{2}$ and mean model size are presented in Table 18 and Table 19. The overall performance of PB-SIS+LASSO, PB-SIS+SCAD and PB-SIS+MCP two-stage approaches for model 5 and model 6 are similar to model 1 to model 4. The PB-SIS+LASSO two-stages approaches with different tuning parameter selection criteria have the highest proportion $P_{2}$ along with largest mean model sizes. On the other hand, the PB-SIS+MCP two stage approaches with different tuning parameter selection criteria end up with the smallest model size on average with a slightly smaller proportion $P_{2}$ than the PB-SIS+LASSO and PB-SIS+SCAD two-stage approaches. Therefore, there is a trade-off between the selection accuracy and the final model size for those two-stage methods. Our suggestion is that we can choose the two-stage PB-SIS+LASSO method when we care more about selecting all true predictors, while the two-stage PB-SIS+MCP approach is a better choice if we would like to find the sparest final model.

We now remark on the choice of a criterion for selecting tuning parameter λ. In the simulations, as mentioned prior to Algorithm 1, one can use cross-validation (CV), AIC [21], BIC [22] or EBIC [23] to choose tuning parameter λ, each of which serves as a model selection criterion. Depending on the property of each model selection criterion, we can choose one for selecting tuning parameter λ based on different needs. CV is a method for choosing a model with the best out-of-sample predictive accuracy. AIC is an efficient model selection criterion, but not consistent. AIC is a method for choosing a model with the minimum disparity between a candidate model and the true model and is very likely to select an overfitted model including more predictors than the true model. BIC is consistent, which means asymptotically BIC chooses the true model. EBIC is extended BIC and consistent as well and may incur a small loss in the positive rate but tightly control the false discovery rate (see [23]). In many applications, CV or BIC is used for selecting tuning parameter λ.

## 5. Application in COPD Gene Expression Data

The simulation studies in Section 4 demonstrate that PB-SIS method can select important variables for generalized linear models with high accuracy rate and low computational cost. We therefore apply the proposed method to a real data example, chronic obstructive pulmonary disease data, which has been utilized in Bahr et al. [24].

Chronic obstructive pulmonary disease (COPD) was classified by the Centers for Disease Control and Prevention in 2014 as the 3rd leading cause of death in the United States (US). COPD weakens lung function and reduces lung capacity. In COPD, there are inflammation of the bronchial tubes (chronic bronchitis) and destruction of the air sacs (emphysema) within the lungs, and the chronic bronchitis and emphysema usually concur under COPD. In addition, the Global Initiative for Chronic Obstructive Lung Disease (GOLD) calls COPD as a common and preventable disease, which is caused by exposure to harmful particles and gases that affect the airways and alveolar of the lungs. The symptoms of COPD include shortness of breath due to lowered concentrations of oxygen in the blood and a chronic cough accompanied by mucus production. COPD progresses with time and the damage caused to the lungs is irreversible.

The main cause of COPD is exposure to tobacco smoke and air pollutants. Problems associated with COPD include under-diagnosis of the disease and an increase in the number of smokers worldwide. Based on previous research, tobacco exposure through smoking cigarettes, second-hand exposure to smoke, continuous exposure to burning fuels, chemicals, polluted air and dust all can cause COPD. Besides tobacco smoke and air pollution, previous study also found that a genetic deficiency, alpha-1 antitrypsin deficiency (AATD), is also associated with COPD. AATD can protect lungs and lungs will become vulnerable due to COPD without AATD. There were over 250 million reported COPD cases in year 2016 and 3.17 million individuals died from this COPD in the year 2015 all over the world. The prevalence of COPD is expected to rise due to increasing smoking rates and aging people in many countries.

Prior to the analysis of the COPD data, we remark on the usage conditions for the difference between the proposed method and some other known methods such as minimum redundancy and maximum relevance (MRMR, e.g., Ding and Peng [25], Radovic et al. [26]) and mutual information feature screening (MIFS, e.g., Hoque et al. [27]). When the response variable of a real data set is binary, the proposed PB-SIS employs point-biserial correlations to conduct feature screening in the first stage and regularization method in the second stage, which ensures that the two-stage variable selection method is consistent. The MRMR method utilizes various measures/criteria (e.g., mutual information difference criteria, mutual information quotient criteria) to maximize relevance and to minimize redundancy and then choose a subset of genes. The MIFS method depends on mutual information and a computational algorithm to obtain a subset of genes. Since the response variable in the COPD data set has two possibles (disease or not disease), it conforms the condition that we use point-biserial correlations for the first-stage feature screening and for GLMs-logit modeling in the second-stage variable selection, so the proposed PB-SIS method is applied to the COPD data for the two-stage feature screening and variable selection. On the contrary to the proposed method, the MRMR and MIFS approaches do not have such restriction on data types or data distributions.

Some previous studies have been conducted for identifying biomarkers for earlier diagnosis of COPD in blood. Ref. [24] compared gene expression profiles of smokers with COPD and smokers without COPD. They applied multiple linear regression to identify candidate genes and pathways.

The goal of our study is to identify disease variability in the gene expression profiles of COPD subjects compared to controls, by re-analyzing pre-existing, publicly available micro-array expression datasets. The data merge resulted in 1262 samples (574 controls and 688 COPD subjects) and 16,237 genes. Our 1262 samples consists of 792 males and 470 females, including 661 former smokers, 418 current smokers and 183 non-smokers.

To check the performance of different variable selection methods, we randomly split the dataset into two parts, the training set and the test set, to evaluate the prediction performance of different methods. The training set contains 80% of the observations and the test set contains 20% of the observations. Thus, the training data sample size is 1010 and the test set sample size is 252. We compare the two-stage PB-SIS approach with the two-stage MMLE and the two-stage Kolmogorov filter approach. For the second stage, we apply three different penalized methods including LASSO [5], SCAD [6] and MCP [17]. For the tuning parameter selection options of each penalized method, we report the results using cross-validation (CV), AIC [21], BIC [22] and EBIC [23].

The final model size and classification accuracy rates are summarized in Table 20. The numbers in the parentheses are the final model size. When we use CV, AIC, BIC, EBIC as the tuning parameter selection criteria, the PB-SIS+LASSO, PB-SIS+SCAD and PB-SIS+MCP methods select a model with higher classification accuracy than the MMLE and Kolmogorov filter with different penalties with the exception of PB-SIS+LASSO with AIC as tuning parameter selection criterion and PB-SIS+MCP with EBIC as tuning parameter selection criterion. When we use more stringent tuning parameter such as EBIC, we can find that the PB-SIS method with different penalties perform significantly better than MMLE with different penalties. For example, when EBIC is used to select tuning parameter, PB-SIS+MCP selects 4 predictors and has a classification accuracy rate 0.817 and the MMLE+MCP method selects 2 predictors and has a classification accuracy rate 0.765. It is clearly demonstrated that by using the two-stage PB-SIS approach, we can select a model with a reasonably good prediction performance and appropriate model size. In Table 20, we bold the best results in each column with a relatively high classification accuracy and a medium model size, indicating that the proposed PB-SIS method and using CV or BIC as the criterion to select tuning parameter can perform best in feature screening and variable selection. Even though using AIC can have a better classification accuracy than BIC, the results have larger model sizes, which is not favorable because AIC tends to select overfitted models.

In the paper of [24], they listed 16 top candidates as the most significant genes in their final selection (Table 2 of the paper). Based on the proposed PB-SIS method, in stage 1, we select 176 genes which have the highest absolute point-biserial correlations with the response variable. However, our selection result does not align with the results in [24]. We judge that this could happen in the gene expression analysis. Both analyses are just exploratory research of the COPD data set, and the real mechanism of COPD is still unknown, so there is no benchmark to compare which selection result is more accurate in reality. Further, no ground truth is available to show which gene does have an association with COPD. So, it is very possible that different approaches can have different results based on different measures. Theoretically, the proposed two-stage PB-SIS method is consistent, which means as the sample size goes to infinity, the procedure selects the true model with probability 1. The simulation results demonstrate that the two-stage PB-SIS method has higher accuracy compared to the MMLE and Kolmogorov Filter approaches in variable selection, and we can select the best model with reasonably good accuracy and appropriate model size in the real data example as in the test set of the simulations. Even though the final gene selection results are not very consistent with the previous study, the proposed method is an effective way for high-dimensional generalized linear model feature screening with high selection accuracy and low computation cost.

## 6. Conclusions and Discussion

We propose a two-stage feature screening method PB-SIS for variable selection of generalized linear models. The point-biserial correlation is utilized as a marginal utility measure to rank and filter the important features that have higher correlation with the response variable in the first stage. After the first stage, the model size can be dramatically reduced from a high-dimensionality *p* to a moderate size *d*. The subsequent step is to further select the important variables and build the final model through a regularization method, such as LASSO, SCAD or MCP. This two-stage approach is confirmed to be very efficient with high selection accuracy and low computational cost.

The PB-SIS method can retain all of the important variables in the selected submodel $M_{d}$ with probability going to one as the sample size goes to infinity. To investigate the performance of the proposed feature screening method, we conduct Monte Carlo simulations. The simulations evaluate the PB-SIS ability for generalized linear models in variable selection by generating data from two different link functions: logit and probit. The simulation results using logit link are presented in this paper. The simulation results using probit link have similar trends, but not presented here. We compare proportion of submodels $M_{d}$ with size *d* that contain all the true predictors among 1000 replications, $P_{1}$, and computing time for our proposed method with the MMLE and Kolmogorov filter methods in three different choices of submodel size *d*. We also compare the MMMS and the associated RSD for those three different feature screening approaches. The simulation results demonstrate that the the proposed method and MMLE perform equally well in almost all settings, but MMLE takes much longer computing time than the proposed method.

The simulation results also show that the proposed method PB-SIS outperforms the Kolmogorov filter method in both selection accuracy and computational cost. We notice that when true predictors have different signs and are highly correlated, all three feature screening approaches fail to select important predictors. Therefore, we need always checking the independence assumption before we apply feature screening approaches. Besides, we also compare the performance of two-stage PB-SIS method with different penalized methods by using different tuning parameter selection criteria. The simulation results show that PB-SIS+LASSO method usually has the highest selection accuracy and the PB-SIS+MCP method can obtain the sparest model.

We also apply the two-stage PB-SIS method to COPD gene expression data. The real data example shows that the PB-SIS method is effective to identify important predictors in the data from the real world.

We comment that the proposed PB-SIS method has limitations. In the application of the proposed method, it is assumed that the response variable is binary data or has a binomial distribution. To achieve a competitive result in variable selection, the proposed PB-SIS method can be applied when the data meets this assumption, and well-performed results are expected. However, if the response variable in a real data set is not binary data, the variable selection result via the proposed PB-SIS method is not an option. In addition, if predictors are not continuous, the result of variable selection using the second stage of the proposed method may be deficient.

Future research are still needed on feature screening for high-dimensional and ultrahigh-dimensional variable selection problems. Even though the PB-SIS method is able to efficiently select important predictors for high-dimensional generalized linear models, it encounters a similar issue as in SIS [9]. Since the PB-SIS method is based on marginal point-biserial correlation ${\hat{\omega }}_{j}$, it tends to miss the important predictors that are marginally uncorrelated but jointly correlated with the response variable. To deal with this issue, Ref. [9] also proposed iterative sure independence screening (ISIS) to use more joint information of the predictors rather than just the marginal information in dimensional variable selection. Therefore, it will be an interesting topic to extend the marginal PB-SIS procedure to an iterative feature screening procedure by iteratively carrying out the marginal screening procedure.

In the numerical studies, we generate predictors from multivariate normal distribution and apply a specific model (generalized linear models) to generate response variable. For future research, we could consider examining the performance of PB-SIS for predictors with heavy tails or outliers. In addition, the proposed method also can be applied in other classical classification methods such as the linear discriminant analysis, quadratic discriminant analysis, robust discriminant analysis or even model-free. Some pioneer work can be found in the related references, including model-free screening procedure for ultrahigh dimensional analysis based on conditional distribution function by [28] and model free feature screening with dependent variables in ultrahigh dimensional binary classification by [29]. 

## Figures and Tables

**Table 1 entropy-25-00851-t001:** Variables included in 6 example models.

Model	Variables	Model	Variables
model 1	$x_{1},x_{3}$	model 4	$x_{1},x_{4},x_{8}$
model 2	$x_{1},x_{6}$	model 5	$x_{1},x_{3},x_{6},x_{10}$
model 3	$x_{1},x_{3},x_{6}$	model 6	$x_{1},x_{4},x_{8},x_{12}$

**Table 2 entropy-25-00851-t002:** The proportion $P_{1}$ and computing time for logit model 1.

			*p* = 200		*p* = 600	
ρ	d	**Method**	$P_{1}$	**Computing Time**	$P_{1}$	**Computing Time**
		PB-SIS	0.995	39.38	0.975	505.59
	⌊n/log(n)⌋	MMLE	0.999	288.42	0.988	1205.60
		Kolmogorov Filter	0.977	48.74	0.919	542.18
		PB-SIS	0.998	38.02	0.988	468.69
0	⌊2n/log(n)⌋	MMLE	1.000	287.70	0.996	1230.62
		Kolmogorov Filter	0.991	47.88	0.962	508.56
		PB-SIS	1.000	39.82	0.992	505.59
	⌊3n/log(n)⌋	MMLE	1.000	283.24	0.998	1243.36
		Kolmogorov Filter	0.996	45.76	0.979	523.13
		PB-SIS	0.995	40.74	0.991	770.15
	⌊n/log(n)⌋	MMLE	1.000	279.03	0.998	1475.47
		Kolmogorov Filter	0.975	50.36	0.957	779.80
		PB-SIS	0.998	42.94	0.988	736.83
0.2	⌊2n/log(n)⌋	MMLE	1.000	261.59	0.999	1428.22
		Kolmogorov Filter	0.993	50.68	0.984	759.09
		PB-SIS	1.000	44.61	0.998	785.37
	⌊3n/log(n)⌋	MMLE	1.000	273.67	0.998	1521.22
		Kolmogorov Filter	0.997	54.36	0.979	787.53
		PB-SIS	0.999	42.84	1.000	567.13
	⌊n/log(n)⌋	MMLE	0.999	282.91	1.000	1263.29
		Kolmogorov Filter	0.997	52.59	0.987	634.73
		PB-SIS	1.000	43.26	1.000	580.38
0.4	⌊2n/log(n)⌋	MMLE	1.000	287.04	1.000	1226.48
		Kolmogorov Filter	0.998	52.45	0.996	583.40
		PB-SIS	1.000	43.59	1.000	558.09
	⌊3n/log(n)⌋	MMLE	1.000	286.37	1.000	1255.63
		Kolmogorov Filter	0.998	50.44	0.998	626.35
		PB-SIS	1.000	42.49	1.000	550.95
	⌊n/log(n)⌋	MMLE	1.000	273.55	1.000	1246.91
		Kolmogorov Filter	1.000	51.20	1.000	549.72
		PB-SIS	1.000	43.03	1.000	546.40
0.6	⌊2n/log(n)⌋	MMLE	1.000	278.38	1.000	1214.87
		Kolmogorov Filter	1.000	49.17	1.000	593.88
		PB-SIS	1.000	44.14	1.000	530.59
	⌊3n/log(n)⌋	MMLE	1.000	290.29	1.000	1268.62
		Kolmogorov Filter	1.000	51.98	1.000	555.51
		PB-SIS	1.000	40.74	1.000	542.07
	⌊n/log(n)⌋	MMLE	1.000	287.68	1.000	1291.00
		Kolmogorov Filter	1.000	51.68	1.000	568.75
		PB-SIS	1.000	42.35	1.000	534.23
0.8	⌊2n/log(n)⌋	MMLE	1.000	287.70	1.000	1230.62
		Kolmogorov Filter	1.000	47.88	1.000	508.56
		PB-SIS	1.000	39.82	1.000	505.59
	⌊3n/log(n)⌋	MMLE	1.000	283.24	1.000	1243.36
		Kolmogorov Filter	1.000	45.76	1.000	523.13

**Table 3 entropy-25-00851-t003:** The proportion $P_{1}$ and computing time for logit model 2.

			*p* = 200		*p* = 600	
ρ	d	**Method**	$P_{1}$	**Computing Time**	$P_{1}$	**Computing Time**
		PB-SIS	0.995	37.37	0.983	437.96
	⌊n/log(n)⌋	MMLE	0.996	268.83	0.984	1184.72
		Kolmogorov Filter	0.978	46.90	0.929	464.81
		PB-SIS	1.000	38.36	0.993	488.97
0	⌊2n/log(n)⌋	MMLE	1.000	274.81	0.994	1185.93
		Kolmogorov Filter	0.994	48.85	0.967	504.09
		PB-SIS	1.000	34.08	0.996	468.94
	⌊3n/log(n)⌋	MMLE	1.000	271.79	0.996	1192.29
		Kolmogorov Filter	0.996	44.15	0.983	478.46
		PB-SIS	0.996	39.26	0.981	755.63
	⌊n/log(n)⌋	MMLE	0.997	273.23	0.982	1489.18
		Kolmogorov Filter	0.980	47.95	0.944	794.48
		PB-SIS	1.000	42.63	0.990	720.63
0.2	⌊2n/log(n)⌋	MMLE	1.000	285.16	0.991	1378.25
		Kolmogorov Filter	1.000	47.37	0.975	717.61
		PB-SIS	1.000	40.98	0.993	723.23
	⌊3n/log(n)⌋	MMLE	1.000	253.00	0.994	1341.77
		Kolmogorov Filter	1.000	46.29	0.985	730.53
		PB-SIS	0.994	41.98	0.977	531.22
	⌊n/log(n)⌋	MMLE	0.994	286.45	0.981	1165.33
		Kolmogorov Filter	0.974	49.49	0.920	537.84
		PB-SIS	0.998	44.08	0.990	532.30
0.4	⌊2n/log(n)⌋	MMLE	0.998	258.44	0.996	1197.96
		Kolmogorov Filter	0.989	44.74	0.956	544.83
		PB-SIS	0.999	37.38	0.999	543.72
	⌊3n/log(n)⌋	MMLE	1.000	245.51	0.998	1251.88
		Kolmogorov Filter	0.994	45.03	0.977	553.20
		PB-SIS	0.970	42.43	0.938	530.02
	⌊n/log(n)⌋	MMLE	0.972	300.79	0.945	1151.30
		Kolmogorov Filter	0.921	49.95	0.839	564.04
		PB-SIS	0.995	40.68	0.976	531.98
0.6	⌊2n/log(n)⌋	MMLE	0.995	268.51	0.978	1188.18
		Kolmogorov Filter	0.968	47.67	0.914	546.76
		PB-SIS	0.997	39.91	0.985	526.89
	⌊n/log(n)⌋	MMLE	0.997	283.26	0.985	1250.08
		Kolmogorov Filter	0.981	47.86	0.951	569.76
		PB-SIS	0.694	39.19	0.514	575.19
	⌊n/log(n)⌋	MMLE	0.684	271.63	0.509	1317.29
		Kolmogorov Filter	0.577	46.40	0.409	541.47
		PB-SIS	0.829	38.37	0.660	537.29
0.8	⌊2n/log(n)⌋	MMLE	0.830	273.97	0.654	1261.09
		Kolmogorov Filter	0.733	51.21	0.571	524.29
		PB-SIS	0.890	40.03	0.729	497.74
	⌊3n/log(n)⌋	MMLE	0.899	272.52	0.731	1319.60
		Kolmogorov Filter	0.855	53.15	0.645	590.93

**Table 4 entropy-25-00851-t004:** The proportion $P_{1}$ and computing time for logit model 3.

			*p* = 200		*p* = 600	
ρ	d	**Method**	$P_{1}$	**Computing Time**	$P_{1}$	**Computing Time**
		PB-SIS	0.935	36.20	0.864	502.72
	⌊n/log(n)⌋	MMLE	0.971	280.09	0.924	1166.15
		Kolmogorov Filter	0.881	52.00	0.740	547.10
		PB-SIS	0.978	38.16	0.922	474.66
0	⌊2n/log(n)⌋	MMLE	0.992	271.48	0.959	1240.70
		Kolmogorov Filter	0.946	46.66	0.825	503.62
		PB-SIS	0.985	39.48	0.943	479.89
	⌊3n/log(n)⌋	MMLE	0.997	276.80	0.970	1136.54
		Kolmogorov Filter	0.965	48.10	0.879	526.78
		PB-SIS	0.961	41.48	0.917	742.75
	⌊n/log(n)⌋	MMLE	0.986	289.34	0.962	1438.22
		Kolmogorov Filter	0.905	55.04	0.798	770.48
		PB-SIS	0.990	42.19	0.967	794.98
0.2	⌊2n/log(n)⌋	MMLE	0.996	277.53	0.988	1466.96
		Kolmogorov Filter	0.959	52.51	0.894	796.17
		PB-SIS	0.992	41.88	0.982	774.20
	⌊3n/log(n)⌋	MMLE	0.998	290.60	0.992	1374.91
		Kolmogorov Filter	0.982	48.75	0.930	733.47
		PB-SIS	0.988	41.07	0.958	565.21
	⌊n/log(n)⌋	MMLE	0.997	279.45	0.975	1248.67
		Kolmogorov Filter	0.950	51.02	0.877	579.06
		PB-SIS	0.997	41.34	0.982	552.32
0.4	⌊2n/log(n)⌋	MMLE	1.000	272.84	0.991	1181.52
		Kolmogorov Filter	0.981	48.93	0.939	578.01
		PB-SIS	0.999	41.43	0.989	525.80
	⌊3n/log(n)⌋	MMLE	1.000	278.69	0.998	1184.50
		Kolmogorov Filter	0.993	50.99	0.961	568.98
		PB-SIS	1.000	40.85	0.994	479.37
	⌊n/log(n)⌋	MMLE	1.000	261.02	0.999	1210.61
		Kolmogorov Filter	0.995	47.03	0.973	539.30
		PB-SIS	1.000	39.03	0.999	521.73
0.6	⌊2n/log(n)⌋	MMLE	1.000	251.97	1.000	1199.08
		Kolmogorov Filter	0.999	55.13	0.990	537.87
		PB-SIS	1.000	39.24	1.000	523.81
	⌊3n/log(n)⌋	MMLE	1.000	301.92	1.000	1161.30
		Kolmogorov Filter	1.000	47.95	0.997	552.51
		PB-SIS	1.000	45.36	1.000	551.47
	⌊n/log(n)⌋	MMLE	1.000	275.32	1.000	1224.08
		Kolmogorov Filter	1.000	48.72	1.000	546.31
		PB-SIS	1.000	39.99	1.000	501.41
0.8	⌊2n/log(n)⌋	MMLE	1.000	274.83	1.000	1268.72
		Kolmogorov Filter	1.000	48.80	1.000	529.60
		PB-SIS	1.000	39.83	1.000	485.13
	⌊3n/log(n)⌋	MMLE	1.000	269.04	1.000	1130.20
		Kolmogorov Filter	1.000	48.96	1.000	537.90

**Table 5 entropy-25-00851-t005:** The proportion $P_{1}$ and computing time for logit model 4.

			*p* = 200		*p* = 600	
ρ	**K**	**Method**	$P_{1}$	**Computing Time**	$P_{1}$	**Computing Time**
		PB-SIS	0.936	35.93	0.883	426.98
	⌊n/log(n)⌋	MMLE	0.940	265.00	0.873	1086.98
		Kolmogorov Filter	0.855	44.41	0.748	465.31
		PB-SIS	0.976	36.54	0.927	443.60
0	⌊2n/log(n)⌋	MMLE	0.977	278.12	0.930	1082.53
		Kolmogorov Filter	0.932	44.96	0.858	465.15
		PB-SIS	0.991	34.45	0.954	433.69
	⌊3n/log(n)⌋	MMLE	0.990	249.25	0.958	1197.08
		Kolmogorov Filter	0.958	46.96	0.900	473.00
		PB-SIS	0.945	39.95	0.851	713.96
	⌊n/log(n)⌋	MMLE	0.949	243.40	0.855	1394.15
		Kolmogorov Filter	0.880	51.41	0.737	712.64
		PB-SIS	0.978	41.57	0.907	802.01
0.2	⌊2n/log(n)⌋	MMLE	0.981	272.46	0.912	1478.99
		Kolmogorov Filter	0.945	49.03	0.833	761.84
		PB-SIS	0.994	46.06	0.936	753.77
	⌊3n/log(n)⌋	MMLE	0.994	274.61	0.938	1431.19
		Kolmogorov Filter	0.969	48.54	0.880	767.63
		PB-SIS	0.909	42.22	0.794	545.33
	⌊n/log(n)⌋	MMLE	0.906	296.03	0.801	1180.15
		Kolmogorov Filter	0.825	50.06	0.657	632.42
		PB-SIS	0.956	42.07	0.881	599.55
0.4	⌊2n/log(n)⌋	MMLE	0.958	285.00	0.882	1280.89
		Kolmogorov Filter	0.922	49.72	0.785	587.09
		PB-SIS	0.980	43.19	0.924	629.28
	⌊3n/log(n)⌋	MMLE	0.980	298.65	0.924	1292.71
		Kolmogorov Filter	0.948	47.11	0.844	629.28
		PB-SIS	0.800	43.01	0.598	525.71
	⌊n/log(n)⌋	MMLE	0.798	276.76	0.588	1280.99
		Kolmogorov Filter	0.635	51.55	0.429	659.16
		PB-SIS	0.896	41.85	0.752	594.56
0.6	⌊2n/log(n)⌋	MMLE	0.904	275.58	0.754	1277.47
		Kolmogorov Filter	0.820	50.37	0.578	579.13
		PB-SIS	0.931	41.78	0.813	545.98
	⌊3n/log(n)⌋	MMLE	0.932	267.36	0.814	1231.71
		Kolmogorov Filter	0.893	53.89	0.684	536.37
		PB-SIS	0.218	46.23	0.059	550.16
	⌊n/log(n)⌋	MMLE	0.216	277.66	0.067	1335.47
		Kolmogorov Filter	0.127	50.44	0.026	554.30
		PB-SIS	0.432	42.89	0.158	526.25
0.8	⌊2n/log(n)⌋	MMLE	0.442	299.96	0.162	1266.48
		Kolmogorov Filter	0.310	56.43	0.099	651.79
		PB-SIS	0.604	44.17	0.264	583.63
	⌊3n/log(n)⌋	MMLE	0.594	278.57	0.270	1247.51
		Kolmogorov Filter	0.463	50.32	0.162	583.90

**Table 6 entropy-25-00851-t006:** The proportion $P_{1}$ and computing time for logit model 5.

			*p* = 200		*p* = 600	
ρ	d	**Method**	$P_{1}$	**Computing Time**	$P_{1}$	**Computing Time**
		PB-SIS	0.844	39.86	0.687	507.72
	⌊n/log(n)⌋	MMLE	0.924	290.59	0.789	1196.92
		Kolmogorov Filter	0.733	47.99	0.477	524.65
		PB-SIS	0.934	45.49	0.806	496.18
0	⌊2n/log(n)⌋	MMLE	0.980	307.30	0.892	1320.07
		Kolmogorov Filter	0.874	48.32	0.660	514.57
		PB-SIS	0.968	38.92	0.865	511.94
	⌊3n/log(n)⌋	MMLE	0.993	281.47	0.923	1203.18
		Kolmogorov Filter	0.925	48.92	0.721	511.44
		PB-SIS	0.872	47.05	0.748	788.45
	⌊n/log(n)⌋	MMLE	0.930	299.69	0.815	1586.81
		Kolmogorov Filter	0.793	53.65	0.510	802.89
		PB-SIS	0.943	46.47	0.840	784.76
0.2	⌊2n/log(n)⌋	MMLE	0.976	292.03	0.920	1580.97
		Kolmogorov Filter	0.885	56.24	0.691	804.61
		PB-SIS	0.967	45.28	0.891	771.98
	⌊3n/log(n)⌋	MMLE	0.990	290.32	0.954	1527.41
		Kolmogorov Filter	0.935	52.22	0.800	806.50
		PB-SIS	0.932	43.66	0.884	574.49
	⌊n/log(n)⌋	MMLE	0.975	291.91	0.923	1304.39
		Kolmogorov Filter	0.868	52.67	0.652	614.94
		PB-SIS	0.983	42.56	0.937	619.95
0.4	⌊2n/log(n)⌋	MMLE	0.994	282.09	0.968	1362.47
		Kolmogorov Filter	0.932	51.89	0.814	672.44
		PB-SIS	0.992	42.75	0.959	628.02
	⌊3n/log(n)⌋	MMLE	1.000	282.06	0.984	1293.01
		Kolmogorov Filter	0.973	51.92	0.904	647.03
		PB-SIS	0.981	44.65	0.956	544.51
	⌊n/log(n)⌋	MMLE	0.994	290.00	0.975	1267.73
		Kolmogorov Filter	0.964	52.80	0.825	580.31
		PB-SIS	0.997	44.76	0.982	553.40
0.6	⌊2n/log(n)⌋	MMLE	1.000	282.58	0.994	1282.41
		Kolmogorov Filter	0.980	52.50	0.925	606.67
		PB-SIS	0.999	45.02	0.989	532.70
	⌊3n/log(n)⌋	MMLE	1.000	285.58	1.000	1222.21
		Kolmogorov Filter	0.989	53.36	0.971	542.50
		PB-SIS	1.000	40.42	1.000	511.58
	⌊n/log(n)⌋	MMLE	1.000	276.40	1.000	1239.96
		Kolmogorov Filter	0.999	52.17	0.990	599.25
		PB-SIS	1.000	42.16	1.000	540.55
0.8	⌊2n/log(n)⌋	MMLE	1.000	277.42	1.000	1187.89
		Kolmogorov Filter	1.000	57.86	0.998	580.64
		PB-SIS	1.000	39.85	1.000	566.37
	⌊3n/log(n)⌋	MMLE	1.000	308.91	1.000	1312.42
		Kolmogorov Filter	1.000	57.67	1.000	587.58

**Table 7 entropy-25-00851-t007:** The proportion $P_{1}$ and computing time for logit model 6.

			*p* = 200		*p* = 600	
ρ	d	**Method**	$P_{1}$	**Computing Time**	$P_{1}$	**Computing Time**
		PB-SIS	0.840	42.99	0.687	466.47
	⌊n/log(n)⌋	MMLE	0.844	291.11	0.693	1189.08
		Kolmogorov Filter	0.745	48.41	0.477	521.30
		PB-SIS	0.935	37.95	0.815	493.00
0	⌊2n/log(n)⌋	MMLE	0.939	270.72	0.824	1183.49
		Kolmogorov Filter	0.872	47.06	0.672	507.86
		PB-SIS	0.960	36.90	0.875	509.89
	⌊3n/log(n)⌋	MMLE	0.964	283.93	0.867	1130.61
		Kolmogorov Filter	0.917	49.45	0.758	489.71
		PB-SIS	0.843	37.87	0.652	716.89
	⌊n/log(n)⌋	MMLE	0.837	268.97	0.656	1429.28
		Kolmogorov Filter	0.708	45.97	0.457	726.55
		PB-SIS	0.929	37.94	0.804	715.15
0.2	⌊2n/log(n)⌋	MMLE	0.930	252.99	0.797	1416.59
		Kolmogorov Filter	0.859	52.56	0.637	739.95
		PB-SIS	0.963	42.50	0.846	746.24
	⌊3n/log(n)⌋	MMLE	0.962	302.94	0.856	1466.25
		Kolmogorov Filter	0.918	54.08	0.738	783.26
		PB-SIS	0.789	41.23	0.583	583.63
	⌊n/log(n)⌋	MMLE	0.795	277.62	0.580	1259.25
		Kolmogorov Filter	0.643	49.71	0.386	605.72
		PB-SIS	0.906	40.48	0.725	606.81
0.4	⌊2n/log(n)⌋	MMLE	0.909	278.11	0.731	1256.01
		Kolmogorov Filter	0.815	54.40	0.578	609.39
		PB-SIS	0.951	41.88	0.644	530.94
	⌊3n/log(n)⌋	MMLE	0.958	282.17	0.802	1309.29
		Kolmogorov Filter	0.892	52.22	0.682	649.21
		PB-SIS	0.600	40.15	0.362	554.15
	⌊n/log(n)⌋	MMLE	0.594	288.81	0.365	1203.03
		Kolmogorov Filter	0.420	49.31	0.184	549.55
		PB-SIS	0.765	40.11	0.544	564.71
0.6	⌊2n/log(n)⌋	MMLE	0.774	264.14	0.540	1235.46
		Kolmogorov Filter	0.670	53.53	0.354	615.18
		PB-SIS	0.849	42.30	0.644	569.00
	⌊3n/log(n)⌋	MMLE	0.849	283.28	0.642	1307.29
		Kolmogorov Filter	0.773	52.34	0.470	621.75
		PB-SIS	0.113	44.71	0.014	558.41
	⌊n/log(n)⌋	MMLE	0.108	284.46	0.016	1166.51
		Kolmogorov Filter	0.051	51.12	0.003	551.09
		PB-SIS	0.319	41.79	0.071	492.50
0.8	⌊2n/log(n)⌋	MMLE	0.318	298.61	0.071	1258.24
		Kolmogorov Filter	0.216	48.99	0.031	561.30
		PB-SIS	0.487	45.32	0.143	527.95
	⌊3n/log(n)⌋	MMLE	0.485	305.90	0.152	1217.13
		Kolmogorov Filter	0.358	52.90	0.082	558.29

**Table 8 entropy-25-00851-t008:** The MMMS and the associated RSD for logit model 1.

		*p* = 200			*p* = 600	
ρ	**PB-SIS**	**MMLE**	**Kolmogorov Filter**	**PB-SIS**	**MMLE**	**Kolmogorov Filter**
0	2 (0)	2 (0)	2 (0.75)	2 (0)	2 (0)	2 (2.24)
0.2	2 (0)	2 (0)	2 (0.75)	2 (0)	2 (0)	2 (0.75)
0.4	2 (0)	2 (0)	2 (0.75)	2 (0)	2 (0)	2 (0.75)
0.6	2 (0.75)	2 (0.75)	2 (0.75)	2 (0.75)	2 (0.75)	2 (0.75)
0.8	3 (0.75)	3 (0.75)	3 (0.75)	2 (0.75)	2 (0.75)	3 (0.75)

**Table 9 entropy-25-00851-t009:** The MMMS and the associated RSD for logit model 2.

		*p* = 200			*p* = 600	
ρ	**PB-SIS**	**MMLE**	**Kolmogorov Filter**	**PB-SIS**	**MMLE**	**Kolmogorov Filter**
0	2 (0)	2 (0)	2 (0.75)	2 (0)	2 (0)	2 (1.68)
0.2	2 (0)	2 (0)	2 (0.75)	2 (0)	2 (0)	2 (1.49)
0.4	2 (0)	2 (0)	2 (1.49)	2 (0.75)	2 (0.75)	2 (2.24)
0.6	3 (1.49)	3 (2.24)	3 (2.99)	3 (2.24)	3 (2.99)	4 (7.46)
0.8	11 (17.91)	11 (16.41)	16 (30.60)	20 (49.25)	20 (47.76)	33 (79.85)

**Table 10 entropy-25-00851-t010:** The MMMS and the associated RSD for logit model 3.

		*p* = 200			*p* = 600	
ρ	**PB-SIS**	**MMLE**	**Kolmogorov Filter**	**PB-SIS**	**MMLE**	**Kolmogorov Filter**
0	3 (1.49)	3 (0.75)	5 (5.22)	4 (5.22)	3 (2.99)	7 (14.93)
0.2	3 (1.49)	3 (0.75)	4 (4.48)	3 (2.99)	3 (1.49)	6 (9.89)
0.4	3 (1.49)	3 (0.75)	4 (2.24)	4 (1.49)	4 (1.49)	5 (5.97)
0.6	5 (1.49)	5 (1.49)	5 (1.49)	5 (1.49)	5 (1.49)	5 (1.49)
0.8	6 (0.75)	6 (0.75)	6 (0.75)	6 (0.75)	6 (0.75)	6 (0.75)

**Table 11 entropy-25-00851-t011:** The MMMS and the associated RSD for logit model 4.

		*p* = 200			*p* = 600	
ρ	**PB-SIS**	**MMLE**	**Kolmogorov Filter**	**PB-SIS**	**MMLE**	**Kolmogorov Filter**
0	3 (1.68)	3 (1.49)	5 (5.22)	4 (3.73)	4 (4.48)	7 (13.43)
0.2	3 (1.49)	3 (1.49)	5 (5.22)	4 (5.22)	4 (5.22)	7 (14.93)
0.4	4 (3.73)	4 (3.73)	6 (8.21)	5 (9.70)	6 (9.70)	11 (23.88)
0.6	8 (9.70)	8 (10.45)	13 (20.90)	14 (28.36)	16 (28.36)	30 (58.21)
0.8	51 (53.17)	51 (51.49)	71 (61.94)	140 (161.94)	140 (160.26)	199 (174.63)

**Table 12 entropy-25-00851-t012:** The MMMS and the associated RSD for logit model 5.

		*p* = 200			*p* = 600	
ρ	**PB-SIS**	**MMLE**	**Kolmogorov Filter**	**PB-SIS**	**MMLE**	**Kolmogorov Filter**
0	6 (6.72)	5 (2.98)	10 (12.69)	10 (20.15)	7 (8.96)	24 (49.25)
0.2	6 (5.22)	5 (2.99)	9 (11.94)	9 (15.67)	7 (8.21)	21 (35.26)
0.4	6 (2.99)	5 (2.99)	8 (7.46)	7 (5.97)	6 (3.73)	13 (17.91)
0.6	7 (2.24)	7 (2.24)	8 (2.99)	8 (2.99)	7 (2.99)	10 (7.46)
0.8	10 (0.75)	10 (0.75)	10 (0.75)	10 (0.75)	9 (0.75)	10 (2.24)

**Table 13 entropy-25-00851-t013:** The MMMS and the associated RSD for logit model 6.

		*p* = 200			*p* = 600	
ρ	**PB-SIS**	**MMLE**	**Kolmogorov Filter**	**PB-SIS**	**MMLE**	**Kolmogorov Filter**
0	6 (6.72)	6 (6.72)	9 (12.69)	10 (17.97)	23 (38.81)	23 (38.81)
0.2	6 (7.46)	6 (6.90)	10 (14.93)	11 (20.90)	11 (20.15)	25 (44.03)
0.4	8 (9.70)	8 (8.96)	14 (18.66)	16 (31.34)	16 (30.22)	32 (56.72)
0.6	16 (23.88)	16 (23.13)	27 (33.58)	37 (64.37)	38 (64.18)	70 (102.43)
0.8	67 (63.43)	68 (61.38)	86 (64.93)	203 (191.04)	200 (191.04)	252 (202.24)

**Table 14 entropy-25-00851-t014:** The proportion $P_{2}$ and mean model size for model 1.

*p*		PB-SIS+LASSO	PB-SIS+SCAD	PB-SIS+MCP
200	CV	0.991 (11.04)	0.989 (5.20)	0.979 (2.80)
	AIC	0.994 (18.74)	0.991 (9.63)	0.990 (8.68)
	BIC	0.992 (12.65)	0.962 (4.69)	0.973 (3.63)
	EBIC	0.985 (14.05)	0.742 (2.07)	0.871 (2.04)
600	CV	0.979 (14.87)	0.979 (7.90)	0.976 (3.68)
	AIC	0.971 (18.31)	0.964 (9.36)	0.962 (8.64)
	BIC	0.968 (15.94)	0.937 (6.54)	0.954 (5.39)
	EBIC	0.960 (16.42)	0.599 (2.31)	0.793 (2.21)

**Table 15 entropy-25-00851-t015:** The proportion $P_{2}$ and mean model size for model 2.

*p*		PB-SIS+LASSO	PB-SIS+SCAD	PB-SIS+MCP
200	CV	0.992 (11.00)	0.993 (5.18)	0.981 (2.80)
	AIC	0.995 (18.79)	0.993 (9.58)	0.994 (8.76)
	BIC	0.993 (12.67)	0.957 (4.85)	0.975 (3.71)
	EBIC	0.978 (14.10)	0.712 (1.98)	0.860 (2.03)
600	CV	0.977 (14.92)	0.975 (7.79)	0.961 (3.65)
	AIC	0.978 (18.26)	0.968 (9.34)	0.967 (8.60)
	BIC	0.975 (15.98)	0.937 (6.52)	0.957 (5.32)
	EBIC	0.973 (16.59)	0.605 (2.43)	0.789 (2.17)

**Table 16 entropy-25-00851-t016:** The proportion $P_{2}$ and mean model size for model 3.

*p*		PB-SIS+LASSO	PB-SIS+SCAD	PB-SIS+MCP
200	CV	0.939 (13.09)	0.933 (6.29)	0.909 (3.85)
	AIC	0.934 (18.36)	0.928 (8.79)	0.925 (8.16)
	BIC	0.932 (15.43)	0.908 (6.63)	0.912 (5.21)
	EBIC	0.929 (17.34)	0.557 (3.25)	0.797 (3.38)
600	CV	0.871 (16.10)	0.871 (8.76)	0.861 (4.68)
	AIC	0.858 (18.11)	0.846 (8.82)	0.846 (8.14)
	BIC	0.858 (17.05)	0.831 (7.46)	0.841 (5.98)
	EBIC	0.856 (18.03)	0.504 (3.55)	0.704 (3.68)

**Table 17 entropy-25-00851-t017:** The proportion $P_{2}$ and mean model size for model 4.

*p*		PB-SIS+LASSO	PB-SIS+SCAD	PB-SIS+MCP
200	CV	0.955 (13.26)	0.953 (6.42)	0.925 (3.89)
	AIC	0.935 (18.36)	0.929 (8.72)	0.926 (8.06)
	BIC	0.931 (15.18)	0.906 (6.52)	0.914 (5.10)
	EBIC	0.927 (17.45)	0.575 (3.26)	0.780 (3.31)
600	CV	0.866 (15.92)	0.865 (8.62)	0.861 (4.68)
	AIC	0.879 (18.10)	0.858 (8.87)	0.846 (8.14)
	BIC	0.878 (17.12)	0.834 (7.40)	0.841 (5.98)
	EBIC	0.880 (18.02)	0.482 (3.50)	0.704 (3.68)

**Table 18 entropy-25-00851-t018:** The proportion $P_{2}$ and mean model size for model 5.

*p*		PB-SIS+LASSO	PB-SIS+SCAD	PB-SIS+MCP
200	CV	0.840 (14.41)	0.838 (7.65)	0.821 (4.89)
	AIC	0.841 (18.45)	0.829 (8.80)	0.827 (8.16)
	BIC	0.838 (16.88)	0.821 (7.50)	0.823 (6.14)
	EBIC	0.838 (18.50)	0.459 (3.33)	0.662 (4.08)
600	CV	0.664 (16.62)	0.661 (9.80)	0.647 (5.69)
	AIC	0.683 (18.26)	0.653 (8.99)	0.655 (8.31)
	BIC	0.684 (17.60)	0.643 (7.89)	0.655 (6.90)
	EBIC	0.684 (18.49)	0.390 (3.55)	0.526 (4.31)

**Table 19 entropy-25-00851-t019:** The proportion $P_{2}$ and mean model size for model 6.

*p*		PB-SIS+LASSO	PB-SIS+SCAD	PB-SIS+MCP
200	CV	0.863 (14.67)	0.859 (7.57)	0.826 (4.94)
	AIC	0.832 (18.30)	0.812 (8.77)	0.811 (8.14)
	BIC	0.830 (16.77)	0.806 (7.47)	0.817 (6.14)
	EBIC	0.833 (18.30)	0.474 (3.45)	0.683 (4.20)
600	CV	0.697 (16.74)	0.696 (9.74)	0.685 (5.77)
	AIC	0.683 (18.24)	0.660 (8.95)	0.653 (8.34)
	BIC	0.683 (17.74)	0.652 (8.03)	0.652 (7.00)
	EBIC	0.684 (18.49)	0.399 (3.56)	0.514 (4.10)

**Table 20 entropy-25-00851-t020:** Two-stage features screening results for COPD gene expression.

		LASSO	SCAD	MCP
PB-SIS	CV	**0.829 (11)**	**0.829 (6)**	**0.829 (6)**
	AIC	0.833 (34)	0.833 (17)	0.833 (17)
	BIC	**0.829 (11)**	**0.829 (6)**	**0.829 (6)**
	EBIC	**0.829 (11)**	0.817 (4)	0.817 (4)
MMLE	CV	0.821 (37)	0.802 (14)	0.806 (5)
	AIC	0.825 (87)	0.806 (37)	0.786 (18)
	BIC	0.817 (20)	0.798 (12)	0.798 (7)
	EBIC	0.802 (11)	0.798 (12)	0.765 (2)
Kolmogorov	CV	0.817 (15)	0.821 (8)	0.790 (3)
Filter	AIC	0.837 (29)	0.786 (14)	0.825 (29)
	BIC	0.821 (3)	0.790 (3)	0.821 (3)
	EBIC	0.821 (3)	0.790 (3)	0.821 (3)

## Data Availability

The data utilized in this study is available and studied in Bahr et al. [24] doi: 10.1165/rcmb.2012-0230OC.

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
