# Peer review of "Feature Screening for High-Dimensional Variable Selection in Generalized Linear Models"

_entropy, 2023, doi:10.3390/e25060851_

Round 1

Reviewer 1 Report

Two stage feature screening method was presented for high dimensional generalized linear models. The results on simulation studies and real example on gene expression data seem to be reasonable and my overall opinion is that this paper is novel and significant and well written. 

It was easy to follow and the method and well presented. 

Author Response

Many thanks to the referee for reviewing this manuscript.

We have proofread the entire manuscript and have corrected the errors which were found in the proofread. We have also improved the exposition. 

Respectfully,
Jinzhu Jiang and Junfeng Shang

Reviewer 2 Report

In this paper, the authors  develop a two-stage feature screening method called point-biserial sure independence screening (PB-SIS) for high-dimensional generalized linear models, aiming for high selection accuracy and low computational cost. I noticed that this method has higher calculation accuracy and speed than other methods listed in the article, and has higher application value. However, there are some concerns need to be fixed. 

1. In the introduction part of the article, some brief introduction about GLM and its applications should be added.

2. I am curious about the usage conditions of the method proposed by the author, and what is the difference between the two-stage feature screening method and some famous feature selection methods (such as: MRMR, MIFS).  which is better? Are our existing data sets strongly assumed to conform to the GLM model if we want to perform the PB-SIS based on a dataset?

3. I note that the authors validated the effectiveness of the proposed method based on COPD gene expression data. I'd be interested to know how this method compares to existing non-GLM model feature selection methods (such as MRMR、MIFS).

4. In Algorithm 1, what principles are generally used to define the value of d? Has the effect of different d values on the feature selection results been verified?

5. What are the limitations of the feature selection method proposed in this paper? Including the assumptions it is based on, the scope of application, etc. If so, please emphasize it at the end of the essay.

Author Response

See attached for the responses.

Reviewer 3 Report

Major remarks:
- Can you indicate which quality assessment method (CV, AIC, BIC, EBIC) gives the best results? From a practical point of view, using all criteria may be difficult.

Minor remarks:
- Can you provide information on where you got the idea for such parameter values in the simulations (section 4.1)?
- I suggest in Table 32 to bold the best results in each column.

Author Response

See attached for the responses.

Reviewer 4 Report

The paper proposes a two-stage feature screening method called point-biserial sure independence screening (PB-SIS) for high-dimensional generalized linear models, especially for logistic models. Through a large set of simulations and one example on gene expression data, the high selection accuracy and low computational cost of PB-SIS are verified. The paper is well-written and it is easy to understand. However, I consider that the paper can be further improved according to the following comments.

1.      The authors employ too much simulations to validate the performance of the proposed method. In my opinion, the methods using probit and logit models to create experimental data are similar since both corresponds to logistic models. At present, the paper includes many tables to summarize the results. I suggest to delete one method to generate the simulated data. Another alternative is to put one method and the corresponding results into supplemental material.

2.      In experiments conducted with simulated data, the number of true important variables is relatively small. I wonder how the methods will perform when the true model includes more variables, such as ten or more ones.

3.      The cited references are relatively old and they should be updated.

4.      In the experiment of gene expression method, the authors only compare the classification performance of the considered methods based on the selected genes. I wonder whether the chosen genes are consistent with those previously found by others to confirm its good discrimination ability.

The English quality of the paper is high and the whole paper is easy to understand. I think the language can be further improved. 

Author Response

See attached for the responses.

Round 2

Reviewer 2 Report

I have no comments.